# Can an Enrichment Programme with Novel Manipulative and Scent Stimuli Change the Behaviour of Zoo-Housed European Wildcats? A Case Study

**DOI:** 10.3390/ani13111762

**Published:** 2023-05-26

**Authors:** Valentina Bertoni, Barbara Regaiolli, Alessandro Cozzi, Stefano Vaglio, Caterina Spiezio

**Affiliations:** 1Research and Conservation Department, Parco Natura Viva—Garda Zoological Park, 37012 Bussolengo, Italy; valentinabertoni4@gmail.com (V.B.); barbara.regaiolli2@unibo.it (B.R.); caterina.spiezio@parconaturaviva.it (C.S.); 2Department of Chemistry, Life Sciences and Environmental Sustainability, University of Parma, 43124 Parma, Italy; 3Department of Biological, Geological and Environmental Sciences, University of Bologna, 40126 Bologna, Italy; 4Research Institute in Semiochemistry and Applied Ethology, 84400 Saint-Saturnin-lès-Apt, France; a.cozzi@group-irsea.com; 5Animal Behaviour and Wildlife Conservation Group, School of Life Sciences, University of Wolverhampton, Wolverhampton WV1 1LY, UK; 6University College—The Castle, Durham University, Durham DH1 3RW, UK

**Keywords:** wildcats, F3 semiochemical, scent enrichment programme

## Abstract

**Simple Summary:**

Enrichment programmes are used to enhance the wellbeing of captive animals in zoos. We aimed to evaluate the effects of an enrichment programme, which included novel objects and scent stimuli, on one group of adult European wildcats hosted at Parco Natura Viva-Garda Zoological Park, Italy. We assessed the behavioural changes following the introduction of novel objects (blocks and rags) and scent (synthetic F3) via observations over four study periods (baseline, rags, F3 rags, blocks). Our results suggest that the enrichment programme did not substantively affect the behaviour of the zoo-housed wildcats but showed potential for both the introduction of novel manipulative objects (rags) and the administration of semiochemicals (F3). However, further work is needed to better assess the effects of the synthetic F3 on welfare in zoo-housed wildcats.

**Abstract:**

Objects and semiochemicals may be used as enrichment in zoos. Domestic cats release Fraction 3 of Facial Pheromone (F3) by rubbing the muzzle to convey relational and territorial information. We aimed to evaluate whether and how the introduction of novel objects and scent stimuli could change the behaviour of one group (N = 5 subjects) of adult European wildcats (*Felis silvestris silvestris*) hosted at Parco Natura Viva-Garda Zoological Park, Italy. We assessed the behavioural changes following the introduction of novel objects (blocks and rags) and scent (synthetic F3) via observations over four experimental conditions (baseline, rags, F3 rags, blocks) using continuous focal animal sampling. Our results showed that no behavioural differences were found between the different conditions and the baseline, except for the condition with blocks when significantly less exploration was observed. Between conditions, wildcats performed significantly less individual explorative, affiliative, and agonistic behaviours, but more individual inactivity, when exposed to rags after F3 administration. Our findings suggest that the enrichment programme did not substantively affect the behaviour of the zoo-housed wildcats. However, the behavioural differences recorded between conditions suggest that, while novel objects introduced as visual stimuli (blocks) do not affect the wildcat behaviour, novel manipulative objects (rags) might impact their behaviour. Moreover, the changes in affiliative and agonistic behaviours displayed during the condition with exposure to rags sprayed with F3 suggest that such semiochemical could play an appeasement role within this study group.

## 1. Introduction

Olfactory stimulation is a form of sensory enrichment that involves placing scents around an enclosure or the presentation of scents to individuals. There is mixed and conflicting evidence regarding the benefits of olfactory enrichment to zoo-housed animal welfare [1,2], but overall it is one of the least studied types of enrichment. However, some studies have shown promise and demonstrated that further work should focus on what constitutes a relevant odour, by considering the ecological/biological relevance of olfactory enrichment to the target species, and how this can be used to create a more stimulating environment for captive animals (e.g., [3,4,5,6]).

Scent-based enrichment offers several benefits for zoo collections. Scents are easy to apply as part of an enrichment programme, either via the introduction of the scents on cloths and other objects, or simply directly placed in the enclosure [2,7,8]. Furthermore, there are a number of different scents that can be introduced, ranging from natural (i.e., prey odours, relevant plant species) and anthropogenic (i.e., essential oils) to fragrances created in laboratories [9,10,11]. For instance, scents from either herbs and plants or animal odours from faeces can be easily obtained in zoos, providing low-priced and varied enrichment [12,13]. Additionally, it is possible to introduce and mix scents in countless ways, thus creating a dynamic enrichment schedule [9].

The effects of scent-based enrichment programmes have been tested by a few studies on domestic, farm, laboratory, and zoo-housed animals (e.g., [14,15]). For example, olfactory enrichment has been studied in both dogs (*Canis lupus familiaris*) and cats (*Felis catus*) housed in rescue shelters as a way to reduce stress and encourage species-specific behaviours (e.g., [8,16]). Specifically, a study of domestic cats [17] found that almost all subjects showed a positive response, including increased play and decreased stress behaviours, when exposed to scented plants, whereas catnip significantly increased play behaviours. 

In zoo-housed animals, olfactory enrichment can be effective at enhancing active and natural behaviours, such as enclosure exploration and scent-marking, ultimately with improved welfare status for the target animals (e.g., [9,10,12,18]). For example, the activity and exploration of black-footed cats (*Felis nigripes*) significantly increased during olfactory enrichment, whilst both resting and standing behaviours decreased [19]. Nevertheless, other studies showed findings that were less clear or even suggested that olfactory enrichment had little effect (e.g., [2,20]).

With regards to designing novel scent-based enrichment programmes, it is crucial to consider both the delivery method and the type of scents used to ensure the effectiveness of the implementation of such programmes [7]. Most studies have used spices or essential oils, rather than focusing on naturally or biologically relevant scents that could be more meaningful to the target species [2]. Many scents, such as lavender, were chosen based on their effectiveness in humans or pets, but these may not necessarily be appropriate for other species [2,21]. However, as with all types of enrichment, the biology of the species should be considered and the effectiveness of any form of putative enrichment should be continually monitored to inform best practices. For instance, several studies have suggested that the use of either natural prey/predator odours or scents from conspecifics should be explored further (e.g., [7,22]). Other studies have advised that the use of diffusers as a delivery method would be more effective (e.g., [7]). Additionally, other authors have indicated that scents could be used in a number of combinations and randomly introduced in order to continue to add novelty to the enrichment programme and so avoid the problem of habituation (e.g., [12]). This could include using synthetic semiochemicals, which have been investigated and have shown some promising potential [23]. 

In felids, synthetic pheromones used in controlled environments derive from the investigation of odour secretions released by domestic cats (*Felis silvestris catus*) and following laboratory resynthesis of the relevant chemical compounds. Specifically, these include two facial semiochemicals, Fraction 3 of facial pheromone (F3) and Fraction 4 of facial pheromone (F4), and a maternal pheromone, the Cat Appeasing Pheromone (CAP) [24]. Facial pheromones are released by the sebaceous glands of the cat’s cheeks. Particularly, domestic cats release F3 by rubbing the muzzle to mark preferred pathways in their territory, or in challenging social environments to facilitate social bonds between conspecifics [24]. In domestic cats, F3 has also been found to play an appeasing function and decrease fear and urine marking behaviours, while promoting exploratory behaviours [25]. Hence, in households and other controlled environments, the synthetic pheromone F3 can be sprayed in specific locations to define comfort sub-areas for the cats, or through the air by diffusers to define larger safety areas [26]. 

European wildcats (*Felis silvestris silvestris*) and domestic cats show a similar behavioural repertoire [27]. Wildcats may therefore be expected to detect and react to synthetic F3 in a similar way to their domestic counterparts. In zoos, F3 has been recently trialled as an environmental enrichment on various feline species (e.g., snow leopards (*Panthera uncia*) and lions (*Panthera leo*); [28,29]) and delivered promising preliminary results to assist with facilitating problematic husbandry and management situations (e.g., abnormal behaviours, veterinary practices) with further cat species (e.g., tigers (*Panthera tigris*) and margays (*Leopardus wiedii*); [30,31,32]).

The aim of this preliminary study is to evaluate the short-term impact of the introduction of an enrichment programme made up of object manipulation, such as rags, sensory enrichment, such as synthetic F3, and novel visual stimuli, such as blocks, on zoo-housed European wildcats by using behavioural observations. If the novel enrichment programme induces positive effects (such as increased affiliative behaviours), the programme could be recommended for further application on zoo-housed felids. Conversely, if negative effects are recorded (such as increased agonistic behaviours, or the performance of abnormal behaviours), further investigation would be necessary to better understand whether it is the overall enrichment programme or a specific stimulus that is responsible for such effects. Finally, whether neutral effects are found, the programme could then be trialled for the management and care of this group of captive European wildcats.

## 2. Materials and Methods

### 2.1. Study Subjects and Housing

This study was conducted over the autumn of 2017 and focused on five adult wildcats (N = 5); a family group including a 10-year-old female and her 5-year-old offspring (two males and two females). The study subjects were neutered to avoid inbreeding and they were healthy without any medical treatment at the time of the study.

The study took place at Parco Natura Viva–Garda Zoological Park, Italy, where the study subjects were born and reared by their parents. Their enclosure measured 64 m^2^ and contained bushes, nest cavities, rocks, and small wooden houses. Wildcats were fed once a day with one fasting day per week and water was provided *ad libitum*. 

Daily environmental enrichment was offered, consisting of manipulative, sensory, and feeding enrichment (see [33] for details on the enrichment plan). Even though the study subjects had prior experience with enrichment procedures, they were naïve to the experiments of this study.

### 2.2. Apparatuses and Experimental Conditions

We provided the study subjects with a novel enrichment programme that included three different types of stimuli: visual and manipulative objects (i.e., rags); visual and manipulative objects sprayed with synthetic F3 (i.e., F3 rags); and new visual objects (i.e., blocks). The first type of stimulus, the visual and manipulative objects, were cloth rags approximately 1-m long (Figure 1). To avoid competition between individuals, we placed five rags in five different points of the enclosure mesh. The rags were knotted to the mesh outside the enclosure and, to keep them attached to the mesh, they were brought back into the enclosure at the end opposite the knot. During the F3 rags condition, to ensure that synthetic F3 spread throughout most of the enclosure, the zookeeper sprayed the rags with F3 (2% active compounds, prepared by the Department of Chemistry at the Research Institute in Semiochemistry and Applied Ethology, Quartier Salignan, France) (Figure 1A). During the F3 rags condition, we provided wildcats with F3 on rags immediately before the beginning of each daily session of behavioural data collection. After that, during the new visual object condition, we used five white blocks hung on the outside of the enclosure mesh in the same positions as the cloth rags to ascertain whether any effect had been due to the presence of such novel objects (Figure 1B). The cats could observe and smell the objects, but they could not reach, manipulate, or play with them.

The stimuli were placed on the first day of each condition and were removed at the end of each condition. However, during the F3 rags condition, the zookeeper knotted the new rags to the mesh and sprayed the F3 on them on the first day, and then sprayed the F3 on the rags already knotted to the mesh every morning until the end of the condition.

The study consisted of four consecutive conditions: -*Baseline* from 11 to 15 September 2017—we did not provide any semiochemical or novel objects to the wildcats.-*Rags* from 18 to 22 September 2017—we provided the wildcats with plain cloth rags.-*F3 rags* from 25 to 29 September 2017—we kept cloth rags and sprayed them with F3.-*Blocks* from 2 to 6 October 2017—we provided the wildcats with plain white blocks.

### 2.3. Data Collection and Analysis

We used continuous focal animal sampling [34] to record the duration of individual and social behaviours during all experimental conditions. Behavioural categories are listed in the ethogram (Table 1). Each condition lasted five consecutive days. For each experimental condition, we carried out ten 20-min sessions (two per day, one session in the morning and one session in the afternoon, consecutively over five days) for each of the five wild cats. Thus, for each condition, we collected behavioural data of the study subjects during 50 sessions (200 sessions over all four conditions) for a total of 4000 min of observations.

During the study period, the zoo was open to the public and the behavioural observations were conducted during opening hours, avoiding the fasting day and the following day. The same observer (V.B.) collected behavioural data throughout the overall study period by carrying out live observations of the study subjects in their social context. Individuals were identified through physical features. The same observer had also collected data on the same study group for our previous study [33].

We defined the daily time budget, both the total during the overall study period and for each condition, to investigate the differences and similarities of the activity budget across the conditions and to assess the behavioural indicators of wellbeing (i.e., positive welfare indicators: species-specific behaviours, meaning the repertoire of behaviours that is shared by virtually all members of a given species; negative welfare indicators: stress-related, self-directed, and abnormal behaviours; [11]) of the study subjects.

We analysed data using non-parametric statistical tests. Specifically, we used the Friedman test to compare the duration of the behavioural categories between the four conditions. As a post hoc test, we used the Nemenyi test because it is a conservative test that was developed to account for a family-wise error [32,35,36]. We set the significance level at *p* < 0.05 and used the statistical software XLSTAT (v. 2020.4.1.1023).

## 3. Results

### 3.1. Time Budget

The most performed behavioural category by the wildcats over the entire study period was individual inactivity (36%), followed by exploration (19%), and social inactivity (11%). Among the other observed behaviours, when considering the overall study period, the least performed behavioural categories were maintenance and agonistic behaviours (around 1% each). Figure 2 shows the total duration as percentages of all the behaviours performed by the study subjects (self-grooming, attention, locomotion, territorial behaviour, social interspecific, not observed). During the entire study period, none of the study subjects displayed abnormal behaviours.

### 3.2. Solitary Behaviours

We compared the durations of individual behaviours of the study subjects during the four conditions. Overall, Friedman tests revealed no significant differences for attention (*Q* = 2.242, *p* = 0.524, *df* = 3), locomotion (*Q* = 7.054, *p* = 0.070, *df* = 3), maintenance (*Q* = 3.596, *p* = 0.309, *df* = 3), territorial behaviours (*Q* = 0.413, *p* = 0.938, *df* = 3), self-grooming (*Q* = 4.401, *p* = 0.221, *df* = 3), and not observed (*Q* = 3.341 *p* = 0.342, *df* = 3) (Figure 3, Table 2). On the other hand, we found a significant difference in the duration of individual inactivity (*Q* = 9.444, *p* = 0.024, *df* = 3) and exploration (*Q* = 27.366, *p* < 0.0001, *df* = 3) between conditions (Figure 3 and Figure 4). Nemenyi post hoc tests revealed that the study subjects displayed significantly less individual inactivity during the rags condition than the blocks condition (*p* = 0.044) (Figure 3); in addition, significantly more exploration was performed during the baseline than during the blocks condition (*p* = 0.001), as well as during the rags than the F3 rags (*p* = 0.019) and blocks (*p* = 0.000) conditions (Figure 4).

### 3.3. Social Behaviours

We compared the durations of the social behaviours of the study subjects during the four conditions. Overall, Friedman tests revealed no significant differences for social interspecific behaviours (*Q* = 3.649, *p* = 0.302, *df* = 3) and social inactivity (*Q* = 0.982, *p* = 0.806, *df* = 3) (Table 2). On the other hand, we found a significant difference between conditions in the duration of affiliative (*Q* = 16.390, *p* = 0.001, *df* = 3) and agonistic behaviours (*Q* = 12.383, *p* = 0.006, *df* = 3) (Figure 5). Nemenyi post hoc tests revealed that the study subjects displayed significantly more affiliative behaviours during the rags than the F3 rags (*p* = 0.032) and blocks (*p* = 0.007) conditions (Figure 5).

## 4. Discussion

This study aimed to assess the effect of an enrichment programme using different novel stimuli (both sensory and manipulative objects) on the behaviour of zoo-housed European wildcats. Enrichment should be designed to trigger naturally occurring behaviours, but it is also important to test a variety of types of enrichment to better understand what works, to avoid unpredictable effects, and to assess the safety of the programme itself [37]. Behaviours can be considered welfare indicators because they reflect an animal’s attempts to cope with the captive environment by indicating situations where welfare is at risk at an early stage [38,39,40,41,42].

Firstly, the time budget results showed that, during the day, the group of zoo-housed European wildcats performed behavioural categories that would be expected in the wild [27,43]. Specifically, individual inactivity was the most performed activity, which was predictable given that felids are often inactive during the day [44], and wildcats are solitary and territorial animals with typical nocturnal/crepuscular activity patterns (usually with peaks at dawn and dusk) [27,43]. This is an elusive species; consistently, the zoo-housed wildcats displayed both territorial behaviour and “not observed” by actively hiding to become non-visible from a visitor’s point of view [27]. The second most performed behaviour was exploration (19% of the total duration of observations), followed by social inactivity (11%), which, together with individual inactivity (36%), counted for 50% of the total time of observations. Interestingly, no abnormal behaviours were observed during the entire study period. The study subjects, therefore, did not show any sign of poor wellbeing or unhealthy conditions, and looked able to cope well with their captive environment.

Overall, the duration of individual activity and exploration, but also of affiliative and agonistic behaviours, showed significant differences across conditions. Particularly, differences between the enrichment conditions and the baseline were not found, except for more exploration between the baseline and blocks conditions, suggesting that only novel visual stimuli would not be effective as enrichment for the wildcats. By contrast, significant differences were found between the conditions with different types of enrichment, suggesting that a variety of enrichments could induce the performance of species-specific behaviours. Rags with synthetic F3 did not elicit any increase in exploratory and affiliative behaviours but induced enhanced inactivity and decreased occurrence of agonistic behaviours. Further behavioural patterns did not change during the introduction of rags, blocks, or F3 rags. 

The largely absent impact of the enrichment programme was surprising. However, we recognize that further factors may have contributed to such a lack of changes induced by the visual and scent enrichments on the wildcat behavioural patterns. For instance, the new familiar scent may not have affected the exploratory behaviour of the study subjects because they could have already established a defined and stable ‘olfactory map’ within their zoo enclosure; i.e., the wildcats were housed in an enclosed zoo environment in which they would not fear predation, challenges to their food supply, etc., so they could lack any need to be ‘hyper-aware’. Moreover, it could also be suggested that objects that can be manipulated may elicit more interest than visual or scent enrichments; i.e., different objects (possibly diffuser blocks or boxes) or scents (other than domestic cat pheromones) could have altered the observations. Additionally, it is important to further reflect on the impact of the design of the experimental conditions; e.g., how the enriching blocks were fixed to the outside of the enclosure (i.e., so that the wildcats could not interact with them), and the enrichment potential of plain rags (i.e., what might it have been about the plain rags that was positive from the wildcats’ perspective).

In domestic cats, synthetic F3 has proven to be successful in promoting exploration [24,25,26]. Moreover, in zoo-housed cat species, F3 has shown potential for enriching various species (e.g., snow leopards and lions; [28,29]), as well as for facilitating complex husbandry and management set-ups for further species (e.g., tigers and margays; [30,31,32]). In addition, our prior study on the behavioural response of zoo-housed wildcats to the reunion of one group member after their long-term separation due to veterinary needs [33] showed that the usage of CAP semiochemical led to minimizing agonistic behaviours and an increased acceptance by the social group. Nevertheless, in our current study on wildcats, we found that exploration was performed more in the presence of plain rags (cloth rags without any added substance) rather than when exposed to rags with F3 or novel visual stimuli, such as blocks. Similarly, affiliative behaviours were performed more during the condition with plain rags than during the F3 rags condition. Thus, in comparison to other novel objects (i.e., blocks) and scents (i.e., F3-scented rags), it seems that the exposure to unscented rags would imply a change in species-specific behaviours displayed by zoo-housed wildcats, promoting active naturally occurring behaviours, such as exploration [45,46,47,48,49] and affiliative social behaviours, with a potentially positive impact on group stability and cohesion [50,51].

However, we acknowledge that the order of the series of enrichment (i.e., unscented rags were always provided to the study subjects after the baseline and before other experimental conditions) may have contributed to the impact of the plain rags; i.e., the most evident response of exploratory behaviour after the presentation of unscented rags may be the effect of novelty, as it was the first stimulus presented. Additionally, the wildcats already had prior experience with enrichment procedures. Thus, the lack of response to stimuli, being significantly different from the baseline, may be the effect of prior experience with other objects; i.e., such a lack of response to stimuli could be the result of generalization or habituation. These factors could partially explain the lack of changes induced by synthetic F3 on exploratory behaviour, while this semiochemical has been previously found to be helpful in problematic scenarios within captive settings (e.g., [32]). However, we also found that individual inactive behaviours were displayed more and, most interestingly, social agonistic behaviours were performed less during the F3 rags condition in comparison to the plain rags and blocks conditions. These results suggest that, when compared with manipulative enrichment and new visual stimuli, F3 scent could affect the behaviour of zoo-housed wildcats and potentially play an appeasement role, as already found for other domestic cat semiochemicals (e.g., [29,33,52,53]). In addition, the reduction of agonistic behaviours with F3-scented rags seems to reaffirm the intent of marking (i.e., social bonding) in wildcats [27,33].

Finally, we must acknowledge some further major limitations of this preliminary study. First, we focused on limited data pools, which included a relatively small sample size and unit of analysis. Moreover, genetics or social learning may have played a role since the study focused on a family group made up of an adult female and her offspring.

## 5. Conclusions

Overall, we can conclude that the newly designed enrichment programme did not fundamentally affect the behaviour of the zoo-housed wildcats. However, our findings suggest that novel manipulative objects might impact wildcat behaviour and synthetic F3 could play an appeasement role in this cat species. Further research should investigate the effect of F3 scent on zoo-house felids, assessing whether and how this semiochemical can influence behavioural indicators of welfare, and if it can be recommended as an effective olfactory enrichment for this species.

## Figures and Tables

**Figure 1 animals-13-01762-f001:**
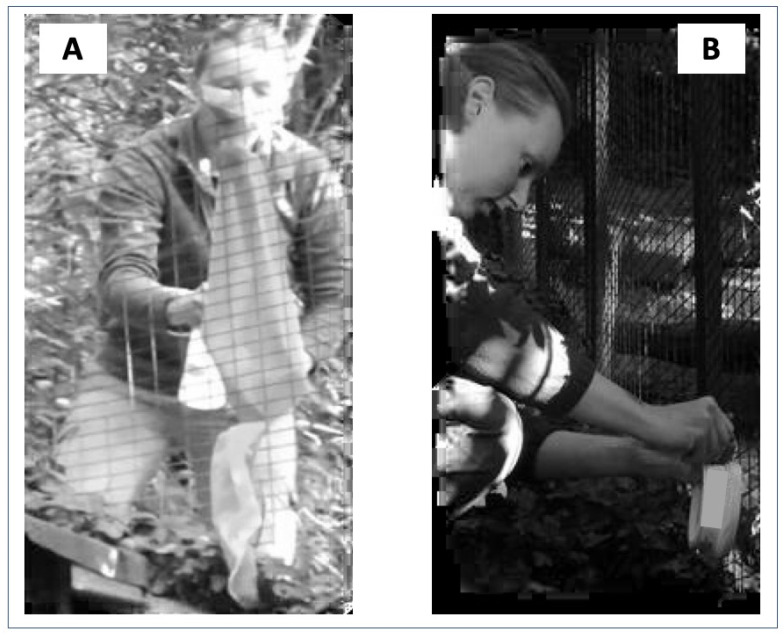
Rags and blocks used in the study. (**A**): Zookeeper spraying F3 synthetic pheromone on a rag fixed on the enclosure mesh during the F3 rags condition. (**B**): Zookeeper fixing a block on the enclosure mesh during the block condition.

**Figure 2 animals-13-01762-f002:**
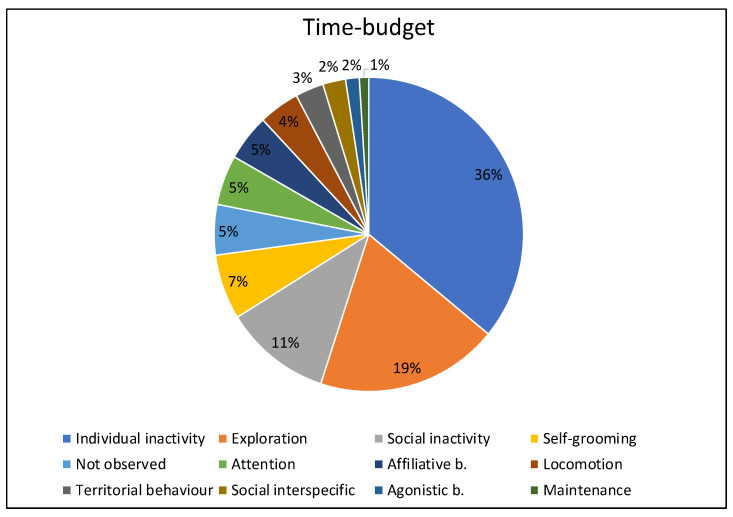
Duration (in %) of behaviours during the overall study period (total duration: 4000 min).

**Figure 3 animals-13-01762-f003:**
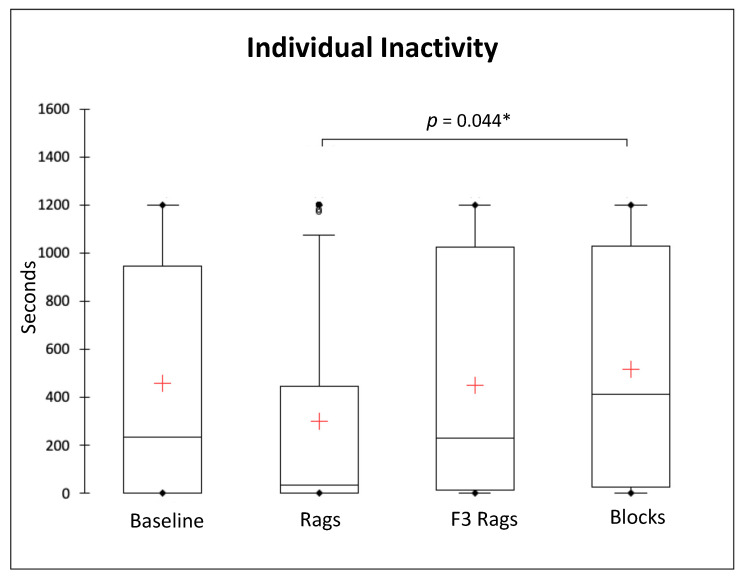
Box and whisker plots of the duration (in seconds) of individual inactivity during different experimental conditions (baseline, rags, F3 rags, blocks). The horizontal line within the box indicates the median, whereas the cross indicates the mean; boundaries of the box indicate the 25th and 75th percentiles, and the whiskers indicate the minimum and maximum values of the data samples. Outliers are drawn as points. Asterisks indicate a significant difference between conditions (Nemenyi post hoc test; *p* < 0.05).

**Figure 4 animals-13-01762-f004:**
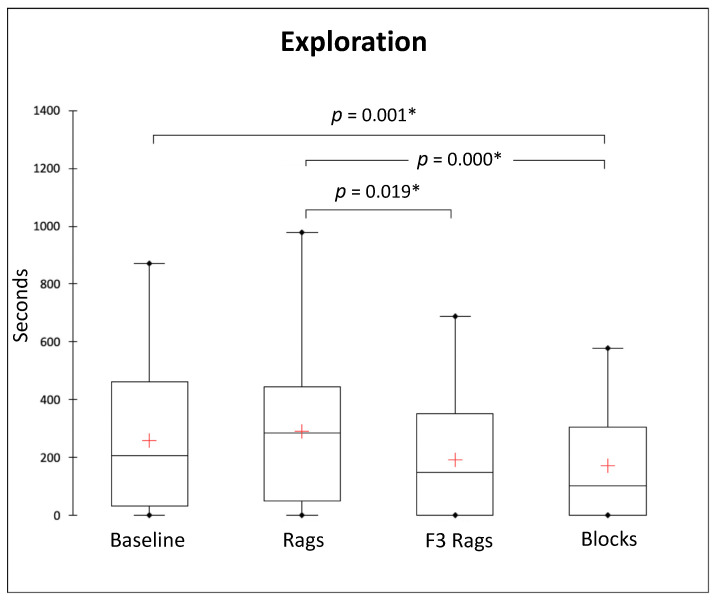
Box and whisker plots of the duration (in seconds) of exploration during different experimental conditions (baseline, rags, F3 rags, blocks). The horizontal line within the box indicates the median, whereas the cross indicates the mean; boundaries of the box indicate the 25th and 75th percentiles, and the whiskers indicate the minimum and maximum values of the data samples. Outliers are drawn as points. Asterisks indicate a significant difference between conditions (Nemenyi post hoc test; *p* < 0.05).

**Figure 5 animals-13-01762-f005:**
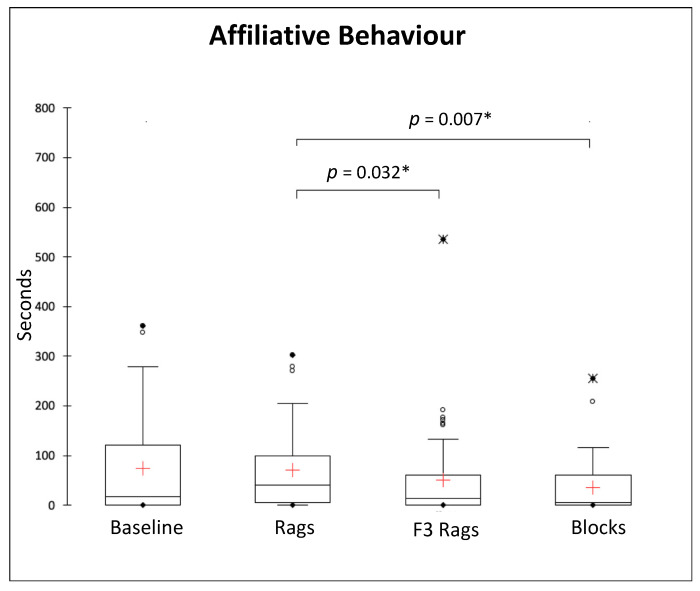
Box and whisker plots of the duration (in seconds) of affiliative behaviours during different experimental conditions (baseline, rags, F3 rags, blocks). The horizontal line within the box indicates the median, whereas the cross indicates the mean; boundaries of the box indicate the 25th and 75th percentiles, the whiskers indicate the minimum and maximum values of the data samples. Outliers are drawn as points. Asterisks indicate a significant difference between conditions (Nemenyi post hoc test; *p* < 0.05).

**Table 1 animals-13-01762-t001:** Ethogram (based on [33], modified).

Behavioural Categories	Definition
Individual behaviours	
Attention	A wildcat is alert and stares at a specific point with straight ears or with ears backwards.
Maintenance	A wildcat defecates and then covers faeces, urinates, eats, and drinks. Include stretching, body shake, individual play, manipulation of plants and other objects, and yawning after waking up.
Exploration	A wildcat visually or olfactorily explores the environment (sniffing the ground or any object).
Self-grooming	A wildcat cleans its fur by licking, scratching, biting, or chewing, or by licking a paw and swiping it on the head with the apparent intent to clean the head.
Locomotion	A wildcat walks, runs, or jumps inside the enclosure.
Territorial behaviours	A wildcat marks the environment by urine spray with a vertical tail and a horizontal urine jet, clawing, rubbing the head against an object, defecating without covering the faces, and patrolling.
Social behaviours	
Affiliative	A wildcat observes, sniffs, or licks another subject or rubs the head and nose against the body of another wildcat.
Agonistic	A wildcat stares at another subject, with ears forward on the head. It can move the tail with fast movements or can have ears flat. Includes agonistic displays such as yawning toward conspecifics, piloerection, raising a paw, and baring teeth.
Social interspecific	A wildcat observes zookeepers, visitors, or animals belonging to species other than its own.
Inactivity	
Individual inactivity	A wildcat sleeps or rests alone.
Social inactivity	A wildcat sleeps or rests in contact with another subject.
Not observed	A wildcat is hiding or is not distinctly visible.

**Table 2 animals-13-01762-t002:** Durations of individual and social behaviours during different experimental conditions. The table reports the median (interquartile range) duration in seconds (s) of the behavioural categories of the study subjects over different conditions (baseline, rags, F3 rags, blocks).

	Baseline	Rags	F3 Rags	Blocks
Individual behaviours				
Attention	29.5 (91.5) s	26.5 (53.5) s	42 (127.5) s	35.5 (149) s
Exploration	205.5 (429.8) s	285 (393) s	149.5 (352.25) s	102.5 (305.25) s
Inactivity	235 (945.5) s	32 (444.25) s	231 (1010.5) s	411.5 (1006.25) s
Locomotion	5.5 (83.25) s	12 (111.75) s	3.5 (77.75) s	0 (68.75) s
Maintenance	7.5 (13.75) s	5 (18) s	3.5 (14) s	3 (11.25) s
Not observed	0 (59.75) s	0 (33) s	0 (69.25) s	7 (54.5) s
Territorial behaviours	0 (5) s	0 (10.5) s	0 (10) s	0 (4) s
Self-grooming	6.5 (55.25) s	19.5 (190.5) s	0 (81.75) s	0 (38.75) s
**Social behaviours**				
Agonistic behaviours	1 (5.5) s	3 (34.75) s	0 (3.5) s	0 (2) s
Affiliative behaviours	17 (120.3) s	40 (94) s	14 (60.75) s	5 (61) s
Social inactivity	0 (0) s	0 (0) s	0 (0) s	0 (0) s
Interspecific	29 (52) s	13 (34.75) s	8 (39) s	17 (51.25) s

## Data Availability

The data presented in this study are available on request from the corresponding author.

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
