# Peer review of "Can an Enrichment Programme with Novel Manipulative and Scent Stimuli Change the Behaviour of Zoo-Housed European Wildcats? A Case Study"

_animals, 2023, doi:10.3390/ani13111762_

Round 1

Reviewer 1 Report

While sample size is stated as a limitation to the study, I am curious whether or not the fact that there is a familial relationship among the cats played a role?  That is, could either genetics and/or social learning have played a role since the subjects were a female and her offspring?  

I, too, was interested in whether or not time of day of the enrichment played a role in the observations.  And I wonder if different scents or objects would have altered the observations.

I find it interesting that agnostic behaviors were reduced with F3 scent - and I think that observation reaffirms the intent of marking (social bonding).

I did find the largely-absent impact of enrichment to be interesting as this almost challenges "common sense".  However, as noted, the cats are in an enclosed zoo environment in which they may not fear predation, challenges to their food supply, etc. so the need to be "hyper-aware" or even "aware" may simply not "be there".

There were several grammatical issues (e.g. "us" instead of "as", line 120; missing a "to" between "enclosure" and "ascertain" in line 156; some odd sentence structure around line 158 and then again around line 180; and the word "Authors" followed by "This" in line 258....) although with a proofread, these are easily remedied.

Reviewer 2 Report

An interesting paper on a not often studied species. The manuscript would benefit from revision, both for clarity in some areas, particularly the methods, and to provide more context for the discussion of the results. 

L175-178 Observation protocol needs clarity. Perhaps remove ‘each period / condition lasted five consecutive days, or move that to L160 where the conditions are described. You state that for each condition you carried out ten observation sessions of each cat, then in the following sentence state that you collected data during fifty sessions for each cat for each condition. There were four experimental conditions, yet L177 states that there were 250 sessions over four periods - 250 is not divisible by 4.  As there were five cats, it may be that “For each experimental condition, ten 20-minute sessions (two per day, over consecutive five days) were carried out for each of the five wild cats” explains what you did. This would give a 50 sessions per condition (ten for each cat), a total of 200 sessions overall.     

L181 – need to define species specific behaviour, backed up with citation for how it’s presence / prevalence can be used to assess welfare

L192-205 Needs simplifying for clarity as it is currently quite hard to follow with the inclusion of reference to the different conditions.

L198 more than 1% could be anything up to 100%.

L207 Fig 2 – pie charts make it very difficult to accurately compare the data presented between charts, and the data are repeated in Tb 2. Consider either presenting a single pie chart (or histogram) of overall activity budget and removing tab 2, or simply keep tab 2. You already have boxplots for the behaviours that were significantly different between conditions.

L210 Rename this section Solitary behaviours. “individual behaviours” gives the impression that you are now looking at all behaviours separately here, when in fact there are more to come in the next section on social behaviours.    

L210-221 (& Social behaviour section) Check Friedman test statistic, it’s more usually χ than Q. An indication of degrees of freedom would be expected to be included.

Figs 3-6 lack titles for the Y axes. Rather than plot the amount of time engaged in the activities, I suggest plotting the percentage or proportion of time (with an indication on the figure as to the total observation period duration). This makes the results more easily comparable between other studies irrespective of their observation duration.

L267 This statement needs backing up with explicit reference made to the time budgets of free-living wild cats

L293-312 Are there studies of other (cat) species that have used F3 or other olfactory enrichment that can be used to give context to your results? Also, the discussion makes very little reference to the significant differences that were found between the conditions.

Whilst the difference between unscented rags and those scented with F3 is considered, it would be good to include a consideration of the conditions, e.g. how enriching blocks fixed to the outside of the enclosure were (i.e. that the cats could not interact with), the enrichment potential of plain rags – what might it have been about the plain rags that was positive from the cat’s perspective, and suggestions for other enrichment possibly blocks (or boxes) in the enclosure.

L37 significant less, correct to significantly less

L143 subjects with and enrichment, correct to subjects with an enrichment

L145 missing full stop after objects

L148 Replace net with mesh for clarity that you mean the enclosure mesh

L153 It would be more usual to refer to experimental conditions rather than periods, e.g. the F3 condition rather than F3 period. This should be done throughout.

L155 we used wight blocks hanged to the mesh, correct to we used five weighted blocks hung on the outside of the enclosure mesh at the same positions as the cloth rags.

L158/9 Last sentence can be deleted following revision above, i.e. remove “Also, in this period the block were five and were 158 hanged in different point of the mesh, at the same side where there were the rags.”

L173 delete ‘method’, continuous focal animal sampling is all that’s required 

L258 Remove ‘Authors’

L266 Group rather than troop?

Reviewer 3 Report

Comments on the manuscript “Can an enrichment programme with novel manipulative and scent stimuli change the behaviour of zoo-housed European wildcats? A case study” submitted to the Animals

General comments

I appreciate the opportunity to review this exciting manuscript.

The study is a case study on the introduction of environmental enrichment to five wild cats. Four types of enrichment were used, one of which had characteristics of olfactory or semiochemical enrichment. Observations lasted 20 minutes for every 10 sessions, two per day. Cats were exposed for five days to each enriching object. The results of comparing baseline with exposure to stimuli do not seem to have been effective. On the other hand, there were behavioral differences between the types of stimuli.

In general, the text is well-written, and has a well-structured logical language. The study is relevant to the well-being of cats in captivity and has a well-structured scientific basis. The Introduction is compelling and well-referenced in the scientific literature.

Despite these qualitative aspects of the text, I have doubts about some sections of the manuscript. I will expose below my doubts and suggestions, when applicable.

ABSTRACT (and conclusion)

The phrase “However, further work is needed to better assess the effects of the

synthetic F3 on behavioural indicators of welfare in zoo-housed wildcats”. This is a conclusion in many papers, but it's a truism that doesn't need to be in the conclusion. Every original article is the product of a part of science and none of them definitively brings a final truth. Every scientific work is open to new experiments, new discoveries, new points of view, etc.

MATERIALS AND METHODS

It is not clear that the cats were naïve to the experiments. It is intriguing that the cats already had prior experience with enrichment procedures (line 305). Lack of response to stimuli significantly different from baseline may be the effect of prior experience with other objects. There is no debate whether the lack of response to stimuli could be the result of generalization or habituation, which are not mutually exclusive. The most evident response of exploratory behavior in the presentation of Rags may be the effect of novelty as it is the first stimulus presented.

Line 142, Apparatuses and study periods: This section of the manuscript remains to be better described. What periods of the day did the observations take place (morning, afternoon, twilight, etc.)? Were the stimuli removed at the end of the sessions? Was there observation on fasting days? Were the cats exposed to public visitation? If there were visitors, how would this change the cats' behavior? Were the cats clinically healthy? What is the reproductive status of cats (neutered or not)?

RESULTS

Line 207: Figure 2 is with low quality. Need to improve the definition of this figure.

Line 234: Confirm that in table 2, the inactivity time in the Rags period is correct.

Line 253: Figure 6 is unnecessary. The agonistic behavior time is minimal and is not informative for the reader in a figure. It is something messy. I suggest just reporting the significant difference in the text (the values are in Table 2).

DISCUSSION

Line 263: What does powerful mean? Doesn't it seem like a vague and subjective term?

Line 316: It is not clear why varying the location of the rags could change the results.

REFERENCES

Line 350: Reference 3 is old. There are other more recent articles to exemplify.

Line 416: Insert the journal (Zoobiology). 

Reviewer 4 Report

The manuscript fits well within the scope of the journal and is of a sound scientific value, making a novel contribution in the field of welfare of zoo animals. Enrichment programmes are important to increase well-being of wild animals kept in zoos. While in some species enrichment programmes are well developed, in others more research is still needed.

The aim of this preliminary study was to evaluate the short-term impact of the introduction of an enrichment programme made up by object manipulation, such as rags, sensory enrichment, such as synthetic F3, and novel visual stimuli, such us blocks, on zoo-housed European wildcats by using behavioural observations. The objectives were clearly defined, the experiment was well designed, relevant references were given and discussed thoroughly.

The limitations of the preliminary study were acknowledged as well as the fact that more research will be needed to fully understand the impact of the tested enrichment on the behaviour of the captive-kept wildcats and to propose an effective scent enrichment for this species. Since this manuscript was clearly identified as a preliminary study, the complex results are not to be expected at this point.

Minor comments:

Figures are quite difficult to read due to their size (especially Figure 2)

Fig. 2 - Durations are given in % but the total duration (100%) is not stated in the legend. The figures need to self-explanatory without the need to look for information elsewhere.

Fig. 3 and following – no indices and units of measurement are given on y-axis nor in the legend
